# Impact of food supplements on early child development in children with moderate acute malnutrition: A randomised 2 x 2 x 3 factorial trial in Burkina Faso

**Mette F. Olsen**[1]*, **Ann-Sophie Iuel-Brockdorff**[1], **Charles W. Yaméogo**[1,2],
**Bernardette Cichon**[1], **Christian Fabiansen**[1], **Suzanne Filteau**[3], **Kevin Phelan**[4],
**Albertine Ouédraogo**[4], **Kim F. Michaelsen**[1], **Melissa Gladstone**[5], **Per Ashorn**[6],
**André Briend**[1,6], **Christian Ritz**[1], **Henrik Friis**[1‡], **Vibeke B. Christensen**[7,8‡]

1 Department of Nutrition, Exercise and Sports, SCIENCE, University of Copenhagen, Copenhagen,
Denmark, 2 Département Biomédical et Santé Publique, Institut de Recherche en Sciences de la Santé,
Ouagadougou, Burkina Faso, 3 Faculty of Epidemiology and Population Health, London School of Hygiene
and Tropical Medicine, London, United Kingdom, 4 The Alliance for International Medical Action (ALIMA),
Paris, France, 5 Department of Women and Children's Health, Institute of Translational Medicine, University
of Liverpool, Liverpool, United Kingdom, 6 Centre for Child Health Research, Tampere University, Faculty of
Medicine and Health Technology, and Tampere University Hospital, Tampere, Finland, 7 Department of
Pediatrics and Adolescent Health, Rigshospitalet, Copenhagen, Denmark, 8 Médecins Sans Frontières-
Denmark, Copenhagen, Denmark

‡ These authors share senior authorship on this work.
* meo@nexs.ku.dk

University Medical School, UGANDA

**Data Availability Statement:** As a university in a
Member State of the European Union, University of
Copenhagen is obliged to comply with the

## Abstract

### Background

Lipid-based nutrient supplements (LNS) and corn–soy blends (CSBs) with varying soy and
milk content are used in treatment of moderate acute malnutrition (MAM). We assessed the
impact of these supplements on child development.

### Methods and findings

We conducted a randomised 2 × 2 × 3 factorial trial to assess the effectiveness of 12
weeks' supplementation with LNS or CSB, with either soy isolate or dehulled soy, and
either 0%, 20%, or 50% of protein from milk, on child development among 6–23-month-
old children with MAM. Recruitment took place at 5 health centres in Province du Passoré,
Burkina Faso between September 2013 and August 2014. The study was fully blinded
with respect to soy quality and milk content, while study participants were not blinded
with respect to matrix. This analysis presents secondary trial outcomes: Gross motor,
fine motor, and language development were assessed using the Malawi Development
Assessment Tool (MDAT). Of 1,609 children enrolled, 54.7% were girls, and median age
was 11.3 months (interquartile range [IQR] 8.2–16.0). Twelve weeks follow-up was com-
pleted by 1,548 (96.2%), and 24 weeks follow-up was completed by 1,503 (93.4%); fol-
low-up was similar between randomised groups. During the study, 4 children died, and
102 children developed severe acute malnutrition (SAM). There was no difference in

provisions of the General Data Protection Regulation. Under Article 9 (2), (j) universities can process sensitive personal data for scientific research purposes. In addition, it is stipulated in Article 9, (4) that Member States may maintain or introduce further conditions, including limitations, with regard to the processing of genetic data, biometric data or data concerning health. The Danish legislation has introduced further conditions in Article 10 of the Danish Act on Data Protection. It is stated in Article 10 of the said Act, that personal research data can be transferred to scientific journals for verification of the research results. However, the Danish Act on Data Protection does not allow for personal data to be made available to others without prior individual approval from the Danish Data Protection Agency. Further information about the requirements of the said Article 10 is available on the University website https://informationssikkerhed.ku.dk/english/protection-of-information-privacy/academic-publications/. The Data Protection Officer of the University of Copenhagen can be contacted about data inquiries at dpo@adm.ku.dk.

**Funding:** The study was funded by Danish International Development Agency (09-097 LIFE) (KFM); Médecins Sans Frontières (Denmark, Norway); Arvid Nilsson's Foundation; The World Food Program, which was part of a donation to the World Food Program from the American people through the support of the US Agency for International Development's Office of Food for Peace; the Alliance for International Medical Action; and the European Union's humanitarian aid funds, in partnership with Action Contre la Faim. The funders had no role in study design, data collection and analysis, decision to publish, or preparation of the manuscript.

**Competing interests:** For other research, KFM has received research grants from US Dairy Export Council and the Danish Dairy Research Foundation, and also has research collaboration with Nutriset, a producer of LNS products, and patent owner; HF has received research grants from ARLA Food for Health Centre, AFH (a consortium between U of Copenhagen, U of Aarhus, and the dairy company ARLA) and Danish Dairy Research Foundation, and also has research collaboration with Nutriset, a producer of LNS products, and patent owner; AB was the inventor of LNS, for which Nutriset has the patent, but abandoned claims to royalties in 2003. Other authors declare no financial relationships with any organisations that might have an interest in the submitted work in the previous three years, and declare no other relationships or activities that

adverse events between randomised groups. At 12 weeks, the mean MDAT z-scores in the whole cohort had increased by 0.33 (95% CI: 0.28, 0.37), $p < 0.001$ for gross motor; 0.26 (0.20, 0.31), $p < 0.001$ for fine motor; and 0.14 (0.09, 0.20), $p < 0.001$ for language development. Children had larger improvement in language z-scores if receiving supplements with milk (20%: 0.09 [−0.01, 0.19], $p = 0.08$ and 50%: 0.11 [0.01, 0.21], $p = 0.02$), although the difference only reached statistical significance for 50% milk. Post hoc analyses suggested that this effect was specific to boys (interaction $p = 0.02$). The fine motor z-scores were also improved in children receiving milk, but only when 20% milk was added to CSB (0.18 [0.03, 0.33], $p = 0.02$). Soy isolate over dehulled soy increased language z-scores by 0.07 (−0.01, 0.15), $p = 0.10$, although not statistically significant. Post hoc analyses suggested that LNS benefited gross motor development among boys more than did CSB (interaction $p = 0.04$). Differences between supplement groups did not persist at 24 weeks, but MDAT z-scores continued to increase post-supplementation. The lack of an unsupplemented control group limits us from determining the overall effects of nutritional supplementation for children with MAM.

## Conclusions

In this study, we found that child development improved during and after supplementation for treatment of MAM. Milk protein was beneficial for language and fine motor development, while suggested benefits related to soy quality and supplement matrix merit further investigation. Supplement-specific effects were not found post-intervention, but z-scores continued to improve, suggesting a sustained overall effect of supplementation.

## Trial registration

ISRCTN42569496.

---

## Author summary

### Why was this study done?

- Moderate acute malnutrition (MAM) affects more than 33 million children globally with long-term consequences for their development.

- There has been a call for research on the effects of MAM treatment on functional outcomes, but previous trials have not assessed effects of the treatments on child development.

### What did the researchers do and find?

- We conducted a randomised trial of 1,609 children with MAM in Burkina Faso to investigate the effectiveness of supplemental foods, including a comparison of matrix (lipid-based nutrient supplements [LNS] or corn–soy blend [CSB]), soy quality (soy isolate or dehulled soy), and content of milk protein (0%, 20%, or 50% of protein).

could appear to have influenced the submitted work.

**Abbreviations:** ALIMA, Alliance for International Medical Action; CRP, C-reactive protein; CSB, corn–soy blend; HAZ, height-for-age z-score; Hb, haemoglobin; ID, identification; IQR, interquartile range; ITT, intention to treat; LNS, lipid-based nutrient supplements; MAM, moderate acute malnutrition; MDAT, Malawi Development Assessment Tool; MUAC, mid-upper arm circumference; NGO, nongovernmental organisation; PP, per protocol; RUTF, ready-to-use therapeutic food; SAM, severe acute malnutrition; WHZ, weight-for-height z-score.

- We assessed child development as gross motor, fine motor, language, and cognitive skills using the Malawi Development Assessment Tool (MDAT).

- We found that child development improved across intervention groups during the supplementation and continued to improve after the supplementation had ended.

- Supplements containing milk protein were beneficial for fine motor and language development.

- Our findings also indicated a benefit of soy isolate over dehulled soy and a benefit of LNS over CSB among boys.

## What do these findings mean?

- Previously reported findings from this trial showed that children mainly put on fat-free mass during supplementation. With the data presented here, we show that motor, language, and cognitive development of children also improve during supplementation and that this improvement is sustained post-supplementation. Our results support the use of milk protein in food products, while suggested benefits of soy isolate and LNS merit further investigation.

- These findings contribute to the needed evidence of supplementation effects on functional outcomes, including child development, and allow better understanding of children's ability to thrive following MAM treatment.

## Introduction

Exposure to malnutrition in early life has a negative impact on the development of motor, language, and cognitive skills in children [1], and the deficits can be tracked through childhood and into adult life [2–4]. Previously, the detrimental effects of malnutrition on brain development were considered structural and irreversible [5,6]. An improved understanding of the plasticity of the brain on top of a humanitarian obligation highlights the importance of treating acute malnutrition and improving its modifiable correlates. Globally, more than 33 million children are affected by moderate acute malnutrition (MAM) [7]. The condition is defined by weight-for-height $z$-score (WHZ) between −2 and −3 and/or a mid-upper arm circumference (MUAC) between 115 and 125 mm [8]. In settings where local foods cannot meet the nutritional needs of children, MAM is treated with food supplements in the form of either fortified blended foods, such as corn–soy blend (CSB) or lipid-based nutrient supplements (LNS) [9], with varying content and quality of protein from soy and milk. There is still limited knowledge of the effects of nutritional supplements for children with MAM. The majority of studies have focused on the impact of supplements on anthropometric outcomes, but recently, there has been a call for evidence of effects on functional outcomes, including child development, to allow a better understanding of children's ability to thrive following MAM treatment [10,11].

We conducted a randomised factorial trial in Burkina Faso in 2013 to 2014 to assess effects of 2 types of food supplements with varying quality of soy and amount of milk protein for treatment of 6 to 23 months old children with MAM. We have previously reported that LNS

compared to CSB improved accretion of fat-free tissue, weight, triceps skinfold, and nutritional recovery rates [12]. In addition, LNS improved haemoglobin (Hb), serum ferritin, and soluble transferrin receptors, although it also increased concentrations of acute phase proteins in children [13]. No consistent effects of soy quality or milk protein was found on these outcomes [12,13]. The overall recovery rate at 3 months was 65% [12]. In this paper, we assess the effectiveness of supplements on motor and language development.

## Methods

### Study setting

As previously described [12], the trial was carried out at 5 health centres in Province du Passoré, Northern Region, Burkina Faso. Study staff were employed at the sites by Alliance for International Medical Action (ALIMA, Senegal). The catchment area covered 143 villages with a total population of approximately 258,000. Province du Passoré is located in one of the most food-insecure parts of the country [14]. At the time of the study, the province was considered a humanitarian setting, qualifying for food supplementation based on a prevalence of general acute malnutrition >10%, while national prevalence was approximately 8% [15]. The main drivers of the food security and nutrition crisis were climate-related shocks, including recurrent droughts, increased food prices, and an influx of refugees after political conflicts in Mali [16]. The high rates of malnutrition among children in the study setting were likely due to inadequate dietary intake and diversity [15] as well as a high prevalence of disease. As we have reported previously, morbidity was very common in the study population with 38% of children recently ill at the time of recruitment and 40% having a positive malaria test on the day of baseline assessment [17]. During the study, ALIMA was implementing a nutrition program in collaboration with local nongovernmental organisations (NGOs) and the Ministry of Health, in which they supported government health structures in the treatment of severe acute malnutrition (SAM). The national guideline for management of malnutrition included treatment of MAM [18], but at the time of the study, the supplementation programme for MAM was functioning only on an irregular basis. Additional details about the study setting have been given previously [12,13,19].

### Study design and participants

The study was part of Treatfood, a randomised $2 \times 2 \times 3$ factorial trial assessing effectiveness of supplement matrix (CSB or LNS), soy quality (dehulled or isolate), and content of milk protein (0%, 20%, or 50% of total protein from dry skimmed milk) in the treatment of MAM (trial registration: ISRCTN42569496). Child development was a secondary outcome of the trial. The sample size calculation was based on the trial's primary outcome (fat-free mass index) [12]. Under the assumption of interaction between factors, a sample of 134 children/arm, or 1,608 in total, would allow detection of a 0.6 SD difference in any pairwise comparison between factors with 80% power, 5% significance level, and allowing for 20% loss to follow-up. In case of no interaction, the sample size would allow detection of smaller effects.

   Prior to the start of the study, community meetings were held in each location to explain the purpose and procedures of the trial to community leaders and other community members. Active screening of children was carried out by community health workers and screening teams in the villages. Children also presented at study sites on caregiver's initiative or were referred from health centres. Eligibility criteria were confirmed MAM diagnosis (MUAC $\geq$115 mm and <125 mm and/or WHZ $\geq$−3 and <−2), age 6 to 23 months, and residency within the catchment area. Children were excluded if they had SAM (WHZ <−3 or MUAC <115 mm or oedema), were already in a nutrition programme, required hospitalisation, or

had been hospitalised within the past 2 months. Children with overt disability, limiting the feasibility of investigations, or with suspected allergies to ingredients in the tested supplements were also excluded.

## Nutritional intervention

All supplements consisted of servings of 500 kcal/day (120 g CSB or 92 g LNS). LNS products were ready to use, while CSB products needed cooking to be consumed as porridge. LNS contained 31.4 to 32.1 g fat and 12.5 to 13.5 g protein per daily serving. CSB contained 11.4 to 11.7 g fat and 15.9 to 16.8 g protein per daily serving. Soy and maltodextrin replaced milk in milk-free products, and more maltodextrin was added when soy isolate replaced dehulled soy. The detailed recipes [12] and micronutrient composition [20] have been published previously. Supplements were manufactured by GC Rieber Compact A/S (Søfteland, Norway).

Caretakers were advised to serve the CSB in 3 meals/day and the sachet of LNS in 1 or more meals/day. If the child was not able or willing to consume the supplements, caretakers were advised to serve smaller and more frequent meals or to mix the supplement in the family foods. Forced feeding was strongly discouraged. Mothers were advised to continue breastfeeding, and family foods were allowed during supplementation. All supplements were introduced as a medical treatment to be exclusively consumed by the child included in the study. Previously reported findings from a qualitative sub-study showed that all supplements were well accepted in the setting [20,21], although CSB products were not consumed as readily as LNS (34% of children were reported to have leftovers of CSB compared to 17% of LNS [20]). LNS products were also more likely to be mixed with other foods and served with a meal or between meals, whereas CSB products were more likely to be served as a separate meal [22].

Children received the allocated supplement for the intervention period of 12 weeks, even if they recovered from MAM at an earlier time point. Children who developed SAM during the intervention period were treated with ready-to-use therapeutic food (RUTF; Plumpy'Nut, Nutriset) instead of the experimental supplement. Children who had not recovered from MAM after 12 weeks of supplementation were provided with RUTF for a maximum of 4 weeks. If still not recovered, they were referred for further clinical examination.

## Randomisation and blinding

At enrolment, children were individually assigned at random to 1 of 12 intervention groups (Fig 1). Allocation followed a random sequence, in blocks of 12 or 24, stratified by study site, prepared using www.randomization.com by a person not otherwise involved in the trial. The supplements were designated by a 1-letter code by the manufacturer. This code was placed in a 12-letter sequence, which was labelled on each supplement during production. The code key was kept in a sealed envelope in a safe until study completion. Only 1 person was aware of the random sequence and code system. This person relabelled supplements with individual study identifications (IDs) and was not otherwise involved in the study. Study participants, outcome assessors, and other study staff were blinded with respect to soy quality, milk content and matrix, while it was not possible to blind study participants with respect to matrix.

## Data collection

Child development was assessed at baseline, at the end of intervention (12 weeks), and post-intervention (24 weeks) using the Malawi Development Assessment Tool (MDAT) at the study sites [23]. Adaptation, validation, and pilot testing of the tool have been reported previously [17]. We assessed the gross motor, fine motor, and language domains of the MDAT. Cognitive functions such as problem-solving, attention, and pre-academic knowledge of

| Matrix | Soy quality | Milk protein% | | |
|---|---|---|---|---|
| | | 0 | 20 | 50 |
| Corn-soy blends | Dehulled | a | b | c |
| | Isolate | d | e | f |
| Lipid-based nutrient supplements | Dehulled | g | h | i |
| | Isolate | j | k | l |

**Fig 1. Experimental food supplements.** The 2 × 2 × 3 factorial design, showing the 12 experimental food supplements based on CSBs or LNS, with either dehulled soy or soy isolate and with 0%, 20%, or 50% of total protein from milk. Two supplements correspond to currently used products: "a" (CSB+) and "b" (CSB++). Product "i" is similar to Plumpy'Sup, containing dehulled soy but with dry skimmed milk instead of whey. CSB, corn–soy blend; LNS, lipid-based nutrient supplements.

numbers and letters are mainly reflected in the tool's language and fine motor domain (e.g., asking the child to find an object under a piece of cloth reflects fine motor as well as problem-solving and memory skills). Due to the young age of participants, we assessed only the first 30 items of each domain [17]. Items were rated as passed (1 point) or failed (0 point). When a child had failed 6 consecutive items within a domain, the assessor would move to the next domain. MDAT assessments were supervised by 2 research assistants with degrees in sociology.

At enrolment, a research nurse collected data on sociodemographic characteristics and 14-day retrospective history of morbidity using a structured questionnaire. Weight was measured in duplicate to the nearest 100 g using electronic scales (Seca model 881 1021659, Hamburg, Germany). Length was measured in duplicate to the nearest 1 mm with a wooden length board. The STATA package "zscore06" was used to calculate WHZ and height-for-age $z$-score (HAZ) [24]. MUAC was measured in duplicate to the nearest 1 mm, at the midpoint between the olecranon and the acromion process of the left arm using a standard measuring tape.

Hb was measured on site using a HemoCue device (Hb 301, Ängelholm, Sweden) [25]. Venous blood was put in a sample tube with clot activator (reference #368492) and transported to the trial laboratory in a cold box at 2 to 8°C. Serum was isolated after centrifugation at 700× g for 5 minutes (EBA 20 S; Hettich, Tuttlingen, Germany) at room temperature and stored at −20°C until shipment to the VitMin Lab in Willstaedt, Germany for analysis of C-reactive protein (CRP) by ELISA [26]. All samples were measured in duplicate, and both intra- and inter-assay coefficients of variation were <10%. Samples were frozen and thawed only once [25].

## Data handling and analysis

Data were double-entered in EpiData 3.1 (EpiData Association, Odense, Denmark), and statistical analyses were carried out using STATA 14.2 (StataCorp, College Station, Texas, United States of America). Descriptive data are shown as mean (SD), median (interquartile range [IQR]), or number (%).

We calculated MDAT z-scores from a Malawian reference population of healthy non-malnourished children [23]. As previously reported [17], we imputed 4 additional items per domain in our data as reference data contained 34 items in each domain. If children had already failed 6 consecutive items during assessment, the values of imputed items were "0." The remaining imputed items were calculated as the mean of the children's last 6 item scores.

These imputed mean values represented 0.2% of all MDAT data at baseline, 0.5% of data at 12 weeks follow-up, and 1.0% of data at 24 weeks follow-up. We conducted a sensitivity analysis, where the imputed items were set as minimum or maximum values, to check the potential impact of imputations on analyses.

Changes in MDAT z-scores during and after supplementation were assessed using paired *t* tests. Effects of the matrix, soy quality, and amount of milk protein on gross motor, fine motor, and language z-scores, respectively, were assessed using linear mixed models. In line with previous reports of trial findings [12,13], the main analysis was intention to treat (ITT) based on all complete cases at the end of supplementation (12 weeks). Effectiveness of supplements were expressed as LNS compared to CSB, soy isolate compared to dehulled soy, and 20% and 50% milk protein compared to 0% milk protein, respectively. Models included adjustment for baseline measure of the outcome, age, sex, MUAC, WHZ, HAZ, month of inclusion, and study site-specific random effects.

We assessed whether potential differences were sustained after supplementation (24 weeks). Models assessing outcomes at 24 weeks were similar to those at 12 weeks but also included interaction between supplement and time as well as participant-specific random effects to capture serial correlations in repeated measurements within children.

For both time points, we carried out model reduction using likelihood ratio tests to compare a model containing the 3-way interactions, corresponding to the $2 \times 2 \times 3$ factorial design, to a model containing only main effects of the factors. If this test indicated any interaction in the factorial layout, we continued by assessing 2-way interactions. When models were reduced to 2-way interactions or main effects, pairwise comparisons of means were performed using model-based post hoc *t* tests. We also considered unadjusted models which only included the intervention groups, baseline outcome measures, and study site-specific random effects (as well as participant-specific random effects for outcomes at 24 weeks). In addition, we conducted per protocol (PP) analyses, including only children who completed 12 weeks of the supplement they were originally allocated to. We used post hoc analyses to explore potential differential effects of supplements between subgroups of children. In these, we assessed effect modification by sex, season, inclusion criteria (WHZ and/or MUAC), stunting ($\geq$ versus $< -2$ HAZ), inflammation (CRP$\leq$ versus $>10$ mg/L), anaemia (Hb$\geq$ versus $<11$ g/dL), and baseline MDAT scores. Model checking was based on residual and normal probability plots. The effects of supplementation are presented in terms of estimated means with 95% CIs. Associations with *p*-values $<0.05$ were considered significant.

### Ethics statement

Caregivers gave informed verbal and written consent (signature or fingerprint) prior to enrolment. Study visits required the presence of children and caregivers during the majority of a day, and they were therefore provided with a meal and activities during the waiting time. Additional incentives, including mosquito nets and soap bars, were given to assure a high follow-up. If participants had not shown up a week after their scheduled study visit, home visits were used to assess the reason for dropout. The trial was approved by the Ethics Committee for Health Research in Burkina Faso (2012-8-059), and consultative approval was obtained from the Danish National Committee on Biomedical Research Ethics (1208204).

### Results

As described previously [12,13], 3,398 children were assessed for eligibility from 9 September 2013 to 29 August 2014, and 1,613 of these were randomly assigned according to the $2 \times 2 \times 3$ factorial design. As 4 children were later excluded as ineligible, a total of 1,609 children were

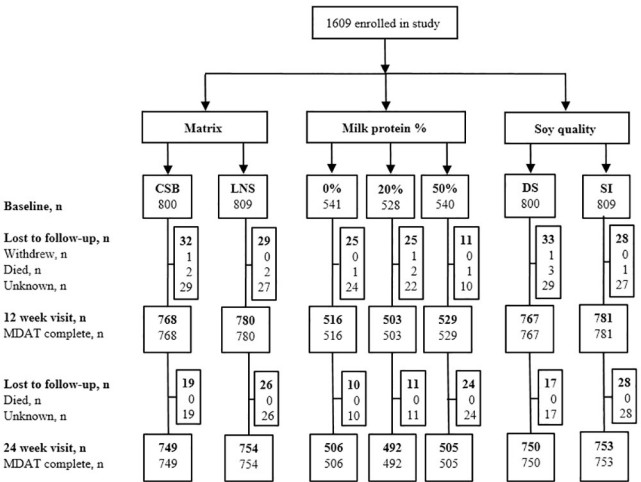

**Fig 2. Trial profile.** CSB, corn–soy blend; DS, dehulled soy; LNS, lipid-based nutrient supplements; MDAT, Malawi Development Assessment Tool; SI, soy isolate.

enrolled in the study (Fig 2). Among the enrolled children, 61 (3.8%) were lost to follow-up at 12 weeks. All of the remaining 1,548 (96.2%) children had complete MDAT data and were included in the ITT analysis of supplementation effects at end of intervention. An additional 45 children (2.8%) were lost to follow-up after supplementation, leaving 1,503 (93.4%) children with complete MDAT data for the ITT analysis of effects post-intervention. Children lost to follow-up had similar baseline characteristics as those with complete data. No children developed allergic reactions to food supplements, but 102 children developed SAM, and 4 children died during supplementation. These adverse events were similar between treatment groups, and none of them were attributed to the food supplements. In the PP analysis, we excluded 119 children (7.7%) children, who received other food during the 12-week intervention because they developed SAM ($n$ = 102) or because of an unconfirmed suspicion of *Salmonella* in 1 of the CSB products ($n$ = 17).

Randomisation resulted in baseline equivalence between intervention groups with regard to MDAT z-scores and key potential confounders (Table 1). At enrolment, children's median age was 11.3 months [IQR 8.2 to 16.0], and 54.7% were girls. Nearly all children (95%) were breastfed at the time of study inclusion. Mean (SD) MDAT z-scores at baseline were −0.39 (1.04) in the gross motor, 0.54 (1.12) in the fine motor, and −0.91 (1.06) in the language domain.

Across all supplementation groups, MDAT z-scores increased during the 12 weeks of supplementation (Table 2). The z-scores continued to increase up to the 24-week visit. The increase of gross motor z-scores was markedly lower after the end of supplementation compared to during supplementation, while language z-scores increased more, but remained the lowest domain z-score at −0.57 (1.91) at 24 weeks.

## Effects of supplements at end of intervention

Milk protein improved language z-scores. Supplements with 20% milk protein increased the mean z-score by 0.09 (95% CI: −0.01, 0.19), although this was not statistically significant, while supplements with 50% milk protein increased the z-score by 0.11 (95% CI: 0.01, 0.21) compared to supplements with 0% milk protein (Table 3). Post hoc analysis showed that this effect

**Table 1. Baseline characteristics of 1,609 children with MAM.**

| | Matrix | | Soy quality | | Milk protein | | |
|---|---|---|---|---|---|---|---|
| | **CSB**<br>*n* = 800 | **LNS**<br>*n* = 809 | **Dehulled**<br>*n* = 800 | **Isolate**<br>*n* = 809 | **0%**<br>*n* = 541 | **20%**<br>*n* = 528 | **50%**<br>*n* = 540 |
| **Sociodemographic characteristics** | | | | | | | |
|  Age, months | 11 [8–16] | 12 [8–16] | 11 [8–16] | 11 [8–16] | 11 [8–16] | 11 [8–16] | 11 [8–16] |
|  Sex, girls | 444 (56) | 435 (54) | 427 (53) | 452 (56) | 295 (55) | 287 (54) | 297 (55) |
| **Breastfed at study inclusion** | 755 (95) | 765 (95) | 754 (95) | 766 (95) | 515 (95) | 493 (93) | 512 (95) |
| **Anthropometry** | | | | | | | |
|  MUAC, mm | 123 (4) | 123 (4) | 123 (4) | 123 (4) | 122 (4) | 123 (4) | 123 (4) |
|  WHZ | −2.2 (0.5) | −2.2 (0.5) | −2.2 (0.5) | −2.2 (0.5) | −2.2 (0.5) | −2.2 (0.5) | −2.2 (0.5) |
|  HAZ | −1.6 (1.1) | −1.7 (1.1) | −1.7 (1.1) | −1.7 (1.1) | −1.7 (1.1) | −1.7 (1.1) | −1.7 (1.2) |
|  Inclusion criteria | | | | | | | |
|  WHZ only | 171 (21) | 166 (21) | 168 (21) | 169 (21) | 111 (21) | 110 (21) | 116 (22) |
|  MUAC only | 225 (28) | 243 (30) | 226 (28) | 242 (30) | 154 (28) | 143 (27) | 171 (32) |
|  WHZ and MUAC | 404 (51) | 400 (49) | 406 (51) | 398 (49) | 276 (51) | 275 (52) | 253 (47) |
| **Hb, g/dL** | 10.0 (1.6) | 10.0 (1.6) | 10.0 (1.6) | 10.1 (1.6) | 10.1 (1.6) | 10.0 (1.7) | 10.0 (1.6) |
| **Morbidity** | | | | | | | |
|  Illness within the last 2 weeks (*n* = 1,599) | 303 (38) | 305 (38) | 318 (40) | 290 (36) | 207 (39) | 206 (39) | 195 (36) |
|  CRP ≥ 10 mg/L (*n* = 1,564) | 188 (24) | 191 (24) | 196 (25) | 183 (23) | 132 (25) | 136 (27) | 111 (21) |
| **MDAT z-scores (*n* = 1,608)** | | | | | | | |
|  Gross motor domain | −0.38 (1.06) | −0.41 (1.02) | −0.40 (1.04) | −0.39 (1.04) | −0.41 (1.03) | −0.39 (0.99) | −0.38 (1.09) |
|  Fine motor domain | 0.54 (1.16) | 0.54 (1.09) | 0.51 (1.11) | 0.56 (1.14) | 0.55 (1.13) | 0.49 (1.07) | 0.58 (1.16) |
|  Language domain | −0.92 (1.10) | −0.90 (1.02) | −0.93 (1.01) | −0.88 (1.10) | −0.88 (1.04) | −0.94 (0.99) | −0.89 (1.14) |

Data are mean (SD), median [IQR], or *n* (%).

CRP, C-reactive protein; CSB, corn–soy blend; HAZ, height-for-age z-score; Hb, haemoglobin; IQR, interquartile range; LNS, lipid-based nutrient supplements; MAM, moderate acute malnutrition; MDAT, Malawi Development Assessment Tool; MUAC, mid-upper arm circumference; SD, standard deviation; WHZ, weight-for-height z-score.

was sex specific (interaction $p$ = 0.02). Among boys, 20% and 50% milk protein increased language z-scores by 0.18 (95% CI: 0.03, 0.32) and 0.18 (95% CI: 0.04, 0.33), respectively. Among girls, no effect of milk protein on language z-scores was seen: 20% (0.01; 95% CI: −0.12, 0.15) and 50% (0.05; 95% CI: −0.08, 0.19). A milk protein content of 20% also increased fine motor z-scores, but only when added to CSB (0.18; 95% CI: 0.03, 0.33). In LNS, the fine motor z-scores were 0.14 lower (95% CI:−0.29, 0.01), but this difference was not statistically significant (interaction between milk and supplement matrix: $p$ = 0.01).

**Table 2. Changes in MDAT z-scores at end of intervention (12 weeks) and after intervention (24 week).**

| Domain | Baseline, 0 weeks | | 12 weeks | | 24 weeks | | Increase: 0–12 weeks | | | Increase: 12–24 weeks | | |
|---|---|---|---|---|---|---|---|---|---|---|---|---|
| | *n* | Mean (SD) | *n* | Mean (SD) | *n* | Mean (SD) | *n* | Mean (95% CI) | *p* | *n* | Mean (95% CI) | *p* |
| **Gross motor** | 1,608 | −0.39 (1.04) | 1,548 | −0,06 (0.96) | 1,503 | 0.07 (0.98) | 1,548 | 0.33 (0.28; 0.37) | <0.001 | 1,489 | 0.13 (0.09; 0.18) | <0.001 |
| **Fine motor** | 1,608 | 0.54 (1.12) | 1,548 | 0.80 (1.00) | 1,503 | 0.93 (1.05) | 1,548 | 0.26 (0.20; 0.31) | <0.001 | 1,489 | 0.13 (0.08; 0.19) | <0.001 |
| **Language** | 1,608 | −0.91 (1.06) | 1,548 | −0.75 (1.06) | 1,503 | −0.57 (1.91) | 1,548 | 0.14 (0.09; 0.20) | <0.001 | 1,489 | 0.19 (0.09; 0.29) | <0.001 |

Differences assessed using paired *t* tests.

CI, confidence interval; MDAT, Malawi Development Assessment Tool; SD, standard deviation.

**Table 3. Effects of supplementary foods on MDAT z-scores at end of intervention (12 weeks, *n* = 1,548).**

| Domain | Matrix | | Soy quality | | Milk protein | | | |
|---|---|---|---|---|---|---|---|---|
| | LNS vs. CSB | *p* | Isolate vs. dehulled | *p* | 20% vs. 0% | *p* | 50% vs. 0% | *p* |
| **Gross motor** | 0.05 (−0.03 to 0.12)[a] | 0.26 | −0.01 (−0.09 to 0.07) | 0.77 | 0.01 (−0.09 to 0.10) | 0.89 | 0.01 (−0.08 to 0.11) | 0.80 |
| **Fine motor** | −0.04 (−0.13 to 0.04) | 0.34 | 0.01 (−0.08 to 0.09) | 0.83 | 0.02 (−0.09 to 0.12)[b] | 0.74 | 0.03 (−0.08 to 0.13)[b] | 0.62 |
| | | | | | In LNS: −0.14 (−0.29 to 0.01) | 0.07 | In LNS: −0.06 (−0.20 to 0.09) | 0.45 |
| | | | | | In CSB: 0.18 (0.03 to 0.33) | 0.02 | In CSB: 0.11 (−0.04 to 0.26) | 0.15 |
| **Language** | 0.001 (−0.08 to 0.08) | 0.97 | 0.07 (−0.01 to 0.15) | 0.097 | 0.09 (−0.01 to 0.19)[c] | 0.08 | 0.11 (0.01; 0.21)[c] | 0.02 |

Data are mean difference (95% CI) based on ITT population. Linear mixed models adjusted for baseline measure of the outcome, WHZ, MUAC, HAZ, age, sex, month of inclusion, and site-specific random effects.

[a]Interaction between sex and matrix shown in post hoc analyses: boys: 0.14 (0.02, 0.25); girls: −0.03 (−0.13, 0.08), interaction *p* = 0.04.

[b]Interaction between matrix and milk protein: *p* = 0.01.

[c]Interaction between sex and milk protein shown in post hoc analyses: boys: 20% milk protein: 0.18 (0.03, 0.32); 50% milk protein: 0.18 (0.04, 0.33); girls: 20% milk protein: 0.01 (−0.12, 0.15); 50% milk protein: 0.05 (−0.08, 0.19), interaction *p* = 0.02.

CI, confidence interval; CSB, corn–soy blend; HAZ, height-for-age z-score; ITT, intention to treat; LNS, lipid-based nutrient supplements; MDAT, Malawi Development Assessment Tool; MUAC, mid-upper arm circumference; WHZ, weight-for-height z-score.

Compared to dehulled soy, supplements with soy isolate increased language z-scores by 0.07 (95% CI: −0.01, 0.15), although this was not statistically significant. No differences between the 2 supplement matrices were seen in the full study population, but post hoc analysis showed a sex-specific effect on gross motor development, in which an effect of LNS compared to CSB was seen in boys (0.14; 95% CI: 0.02, 0.25), but not in girls (−0.03; 95% CI: −0.13, 0.08, interaction *p* = 0.04).

The effects of supplements were not modified by season, MAM-defining criteria (WHZ or MUAC), stunting, inflammation, anaemia, or MDAT scores at baseline. We found similar results if analyses were adjusted only for baseline measure of the outcome and study site (Table A in S1 Table) or in the PP analysis (Table B in S1 Table). Sensitivity analyses showed that the assessment of intervention effects was unaffected by imputation of MDAT items (Tables C and D in S1 Table).

## Effects of supplements post-intervention

At the 24-week visit, there were no differences in MDAT z-scores between supplementation groups (Table 4), and there were no differential effects of supplements between boys and girls.

**Table 4. Effects of supplementary foods on MDAT z-scores after intervention (24 weeks, *n* = 1,503).**

| Domain | Matrix | | Soy quality | | Milk protein | | | |
|---|---|---|---|---|---|---|---|---|
| | LNS vs. CSB | *p* | Isolate vs. dehulled | *p* | 20% vs. 0% | *p* | 50% vs. 0% | *p* |
| **Gross motor** | −0.004 (−0.09; 0.08) | 0.92 | 0.01 (−0.07; 0.09) | 0.84 | −0.04 (−0.14; 0.06) | 0.40 | −0.05 (−0.15; 0.05) | 0.33 |
| **Fine motor** | 0.01 (−0.07; 0.10) | 0.75 | 0.03 (−0.06; 0.11) | 0.58 | 0.01 (−0.10; 0.12) | 0.81 | −0.04 (−0.15; 0.07) | 0.51 |
| **Language** | −0.08 (−0.21; 0.04) | 0.20 | 0.08 (−0.04; 0.21) | 0.18 | −0.05 (−0.20; 0.11) | 0.55 | −0.02 (−0.17; 0.13) | 0.79 |

Data are mean difference (95% CI) based on ITT population. Linear mixed models are adjusted for baseline measure of the outcome, WHZ, MUAC, HAZ, age, sex, and month of inclusion. The models include interaction between supplement and time and both site- and participant-specific random effects. There were no interactions between the 3 factors (all *p* > 0.28).

CI, confidence interval; CSB, corn–soy blend; HAZ, height-for-age z-score; ITT, intention to treat; LNS, lipid-based nutrient supplements; MDAT, Malawi Development Assessment Tool; MUAC, mid-upper arm circumference; WHZ, weight-for-height z-score.

Results were similar in analyses with adjustment only for baseline measure and study site (Table E in S1 Table) and in the PP population (Table F in S1 Table).

## Discussion

We have shown that children with MAM improved their motor, language, and cognitive development during and after 12 weeks of food supplementation. We found that supplements with milk protein appeared to be more effective in improving fine motor skills, when added to CSB, and language skills, regardless of supplement matrix. We also found that soy isolate over dehulled soy increased language z-scores, although the difference was not statistically significant. Post hoc subgroup analyses showed sex-specific effects of supplementation as the effect of milk protein on language development was found only in boys, and boys also had a beneficial effect of LNS over CSB on gross motor development. The differences between supplementation groups were not sustained 12 weeks post-supplementation, but children continued to increase their z-scores in all development domains.

The main strengths of our study are the randomised factorial design and the large study sample with low attrition and high degree of data completeness. Furthermore, we assessed motor and language development using a well-validated tool with high inter- and intra-rater reliability and sensitivity in our study setting [17] and predictive validity for later school performance shown from other settings [27]. It is also a strength that our study included analyses of effects by sex and whether effects were sustained post-supplementation, which are rarely reported from nutritional trials. However, the sex-specific effects should be interpreted with caution, as subgroup analyses were added post hoc. During supplementation, 6.3% of children in our study deteriorated to SAM, which is similar to the rates reported from other studies among children with MAM [10]. Our study did not include an unsupplemented control group, as treatment of MAM was already on the national agenda in Burkina Faso at the time of the study. This limits us from concluding on the overall effects of nutritional supplementation for children with MAM. The observed increase in development scores across intervention groups might be partly attributed to learning effects of the MDAT tool, although the study does not allow us to quantify these. As in other food intervention trials, the interpretation of our findings is also limited by the fact that the content of some ingredients varied between products to compensate for the differences between the experimental factors. Consequently, the shown effects of supplements with milk are in comparison to supplements without milk, but with more soy and maltodextrin, which were added to keep the amount of macronutrients and energy constant. Lastly, the interpretation of effects of supplements is limited by lack of detailed information about individual adherence to the intervention as it is not feasible to collect this type of data in a large sample of children. Lastly, the nature of our trial did not allow us to conclude on the underlying mechanisms of the effects of supplementation. The effects on language and motor skills may be partly related to increased well-being related to less hunger and malnutrition as a result of providing a supplement. The use of supplementation in our study is likely to be similar to the actual use in a nutritional programme, and the observed effects are therefore highly relevant for practice.

According to recent systematic reviews of treatment for MAM [10,11], no previous trials have assessed effects on child development. However, a number of studies have assessed effects of nutritional interventions on child development in low- and middle-income settings. A Cochrane review from 2019 assessed effects of LNS in non-malnourished children and found that 11 trials had assessed effects on child development outcomes [28]. The results could not be combined due to differences in outcomes and control interventions, but the majority of studies found effects on 1 or several psychomotor and neurodevelopment tests. Two systematic

reviews from 2015 included studies among children below 2 years in low- and middle income settings ranging from well nourished to moderately malnourished [29,30]. Both reviews showed that most nutritional trials were underpowered to detect effects on child development scores. In meta-analyses of 18 and 23 studies, respectively, pooled effect estimates of nutritional supplements on development scores were significant with small effect sizes of 0.09 SD (95% CI: 0.03 to 0.14) and 0.08 SD (95% CI: 0.02 to 0.13). Most of the reviewed interventions were micronutrient supplements. A subgroup of 7 studies (893 participants) providing macronutrients showed a slightly larger effect size of 0.14 SD (95% CI: 0.01 to 0.27) [30]. The reviewed studies tested a variety of food supplements against controls receiving little or no energy. In comparison, the effects we report of milk protein correspond to 0.10 to 0.17 SD and the effects of LNS over CSB in boys correspond to 0.14 SD. In terms of clinical relevance, we note that these effect sizes correspond to the differences seen between children with and without anaemia at baseline [17]. The total increase across supplementation groups was 0.13 to 0.33 SD at 12 weeks and 0.24 to 0.46 SD at 24 weeks.

To the best of our knowledge, none of these previous studies have investigated whether effects of food supplementation differed between girls and boys. However, sex-specific effects among children have been reported for various other interventions, including vaccines [31], vitamin A [32], maternal vitamins [33], school meals [34], and fatty fish consumption [35]. The patterns of these subgroup effects are often complex, and their mechanisms are not well understood. It is likely that genetic and hormonal differences affect both biochemical brain processes and behaviour differently in boys and girls. The sex-specific effects indicated in our post hoc assessment merit further investigation, and we suggest that future supplementation trials include a prespecified subgroup analysis by sex.

The gross motor domain of MDAT represents children's ability to crawl, stand, and walk, thus the effect of LNS over CSB seen in boys may be linked to the trial's primary finding that LNS was beneficial for accretion of fat-free mass [12]. In contrast, the effects of milk protein were seen in the tool's fine motor and language domains, which also reflect cognitive functions such as problem-solving, reasoning, and executive functions. Milk products have been used in food supplementation since early programme initiatives [36] and have been studied for their benefits for growth and health, especially in children with poor nutritional status [37]. However, very few studies have assessed the effects of milk on cognitive outcomes. A study in Ghana provided school children with 8.8 g milk protein in daily school meals for 4.5 months and showed effects on 2 of 5 cognitive tests at the end of supplementation, but did not report if these effects were sustained at a 9-month assessment [38]. Among school children in Kenya, supplements containing either meat or milk resulted in higher school end-term test scores, while only children receiving meat improved the cognitive measures tested in the study [39,40]. As reported previously, children in our study had a low intake of animal-source foods [41], but 95% of children were still breastfed at the time of study inclusion. Bioactive peptides from milk proteins, such as α-lactalbumin, may benefit cognition [42], but it is unlikely that these have caused the milk effects seen in this study, where the majority were breastfed, given that the content of α-lactalbumin in human milk is much higher than in cow's milk [43]. The effects of providing children with milk protein might be different in populations where the usual dietary intake includes more or less animal-source foods or where fewer children are breastfed.

The effects of milk over soy are not necessarily caused by protein quality, but perhaps by accompanying nutrients. As discussed previously [12], milk was added as skimmed milk powder, which contains lactose and minerals with high bioavailability. Vitamin B-12 in supplements came from the vitamin premix as well as dry skimmed milk, and B-12 levels were 10% and 25% higher in products with 20% and 50% milk protein, respectively, compared to

products without milk. Thus, a higher intake of vitamin B-12 may have contributed to cognitive effects in children receiving supplements with milk [44,45].

There is also a possibility that the observed effects of milk protein reflect adverse effects of isoflavones found in soy [46]. Although we did not directly compare milk versus soy, products with less milk protein contained more soy flour. Isoflavones, in particular genistein, are suspected to negatively impact growth and cognitive development through effects on iodine uptake by the thyroid gland [47]. Thus, it might be speculated whether isoflavones could explain both milk effects and the (statistically insignificant) benefit of soy isolate over dehulled soy. However, the goitrogen effect of soy may occur if iodine intake is marginal, which was not the case in our study setting, which has a successful salt iodisation programme [48]. Lastly, while we are not able to explain why the effects of milk on fine motor development were only seen when added to CSB products, we speculate that the additional benefit of milk may not be observable in the more nutrient-dense LNS matrix.

The supplement-specific effects were not sustained at the 24-week follow-up. Children who had not recovered from MAM at end of 12-week supplementation were referred to additional LNS in the form of RUTF. Since nutritional recovery was higher in the LNS group compared to the CSB group [12], we expect that more children from the CSB group received post-intervention RUTF, which may have caused some dilution of intervention effects. Additional supplements or treatments that may have been given post-intervention were beyond the control of our trial, but we have no reason to suspect that these differed among intervention groups.

Since all groups were supplemented, it is not possible to assess whether an overall effect of supplementation for children with MAM was sustained post-intervention. However, we note that the increase in age-adjusted z-scores during and after supplementation were larger than any differences between intervention groups and find it likely that the main long-term benefit to development is from getting a supplement regardless of its matrix or protein type. There has been a concern that the use of food supplements for treatment of MAM might contribute to the double burden of malnutrition if children accumulate too much fat mass [49]. However, as we have previously shown from this trial, children put on mainly fat-free mass during supplementation [12]. With the child development data presented here, we show that motor, language, and cognitive development of children also improve during supplementation and that this improvement is sustained post-supplementation.

Finally, it is important to note that impaired development is not just a consequence of inadequate diet. Other aspects of poverty exacerbate the negative influence of malnutrition and interact in their effects on child development [5,6,50]. Malnutrition has indirect effects on cognitive development through decreased interactions and withdrawing from peers and surroundings [50]. We do not have information about responsive caregiving or stimulation in the households of our study sample, but it is likely that this could protect against the impact of malnutrition or that both adequate nutrition and stimulation are needed for the child to develop to its full potential [51]. While the social and economic aspects of a child's environment are not easily changed, providing adequate nutrition during infancy and later will at least lessen the cognitive deficits of poverty.

In conclusion, the improved child development scores seen in this study support the use of food supplements for children with MAM. Our results also support the inclusion of milk protein in food supplements to benefit children's fine motor, language, and cognitive development, while the benefits relating to product type and soy quality were less clear. The differences between groups were not sustained after supplementation was discontinued, but both motor and language development scores continued to increase in all groups. Further investigation should look into sex-specific effects of supplementation and additional effects of improving responsive caregiving stimulation of children.

## Supporting information

**S1 Table. Supplementary tables.** Table A in S1 Table: effects of supplementary foods on MDAT z-scores at end of intervention (12 weeks, *n* = 1,548): unadjusted analysis. Table B in S1 Table: effects of supplementary foods on MDAT z-scores at end of intervention (12 weeks) in PP population (*n* = 1,548). Table C in S1 Table: effects of supplementary foods on MDAT z-scores at end of intervention (12 weeks, *n* = 1,548): sensitivity analysis using minimum values for data imputation. Table D in S1 Table: effects of supplementary foods on MDAT z-scores at end of intervention (12 weeks, *n* = 1,548): sensitivity analysis using maximum values for data imputation. Table E in S1 Table: effects of supplementary foods on MDAT z-scores after intervention (24 weeks, *n* = 1,503): unadjusted analysis. Table F in S1 Table: effects of supplementary foods on MDAT z-scores after intervention (24 weeks) in PP population (*n* = 1,382). MDAT, Malawi Development Assessment Tool; PP, per protocol.
(DOCX)

**S1 Text. CONSORT checklist.** CONSORT, Consolidated Standards of Reporting Trials.
(DOC)

**S2 Text. Treatfood study protocol.**
(PDF)

## Acknowledgments

We acknowledge the contributions of the study participants and their families, the research staff, the Ministry of Health in Burkina Faso, the health and village authorities in Province du Passoré, and the staff at the health centres.

## Author Contributions

**Conceptualization:** Suzanne Filteau, Kim F. Michaelsen, Per Ashorn, André Briend, Henrik Friis, Vibeke B. Christensen.

**Data curation:** Ann-Sophie Iuel-Brockdorff, Charles W. Yaméogo, Bernardette Cichon, Christian Fabiansen, Albertine Ouédraogo.

**Formal analysis:** Mette F. Olsen, Christian Fabiansen, Melissa Gladstone, Christian Ritz, Henrik Friis, Vibeke B. Christensen.

**Investigation:** Ann-Sophie Iuel-Brockdorff, Charles W. Yaméogo, Bernardette Cichon, Christian Fabiansen.

**Methodology:** Mette F. Olsen, Ann-Sophie Iuel-Brockdorff, Suzanne Filteau, Melissa Gladstone.

**Project administration:** Ann-Sophie Iuel-Brockdorff, Charles W. Yaméogo, Bernardette Cichon, Christian Fabiansen.

**Writing – original draft:** Mette F. Olsen.

**Writing – review & editing:** Mette F. Olsen, Ann-Sophie Iuel-Brockdorff, Charles W. Yaméogo, Bernardette Cichon, Christian Fabiansen, Suzanne Filteau, Kevin Phelan, Albertine Ouédraogo, Kim F. Michaelsen, Melissa Gladstone, Per Ashorn, André Briend, Christian Ritz, Henrik Friis, Vibeke B. Christensen.

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
