## [Editor Report · Decision Letter 0]

8 Jun 2020

Dear Dr Olsen, 

Thank you for submitting your manuscript entitled "Effectiveness of food supplements on early child development in children with moderate acute malnutrition: a randomized 2 x 2 x 3 factorial trial in Burkina Faso" for consideration by PLOS Medicine.

Your manuscript has now been evaluated by the PLOS Medicine editorial staff as well as by an academic editor with relevant expertise and I am writing to let you know that we would like to send your submission out for external peer review.

Kind regards,

Artur Arikainen,

Associate Editor

PLOS Medicine

---

## [Decision Letter · Decision Letter 1]

28 Jul 2020

Dear Dr. Olsen,

Thank you very much for submitting your manuscript "Effectiveness of food supplements on early child development in children with moderate acute malnutrition: a randomized 2 x 2 x 3 factorial trial in Burkina Faso" (PMEDICINE-D-20-02543R1) for consideration at PLOS Medicine. 

Your paper was evaluated by a senior editor and discussed among all the editors here. It was also discussed with an academic editor with relevant expertise, and sent to three independent reviewers, including a statistical reviewer (reviewer #2). The reviews are appended at the bottom of this email and any accompanying reviewer attachments can be seen via the link below:

[LINK]

In light of these reviews, I am afraid that we will not be able to accept the manuscript for publication in the journal in its current form, but we would like to consider a revised version that addresses the reviewers' and editors' comments. Obviously we cannot make any decision about publication until we have seen the revised manuscript and your response, and we plan to seek re-review by one or more of the reviewers. 

We expect to receive your revised manuscript by Aug 18 2020 11:59PM. Please email us (plosmedicine@plos.org) if you have any questions or concerns.

We look forward to receiving your revised manuscript. 

Sincerely,

Emma Veitch, PhD

PLOS Medicine

On behalf of Clare Stone, PhD, Acting Chief Editor,

PLOS Medicine

plosmedicine.org

*Per the journal's usual guidelines, we'd ask that the authors include in the Abstract Methods and Findings section a brief summary of any key limitation(s) of the study's methodology.

*At this stage, we ask that you include a short, non-technical Author Summary of your research to make findings accessible to a wide audience that includes both scientists and non-scientists. The Author Summary should immediately follow the Abstract in your revised manuscript. This text is subject to editorial change and should be distinct from the scientific abstract. Please see our author guidelines for more information: https://journals.plos.org/plosmedicine/s/revising-your-manuscript#loc-author-summary

*It's not clear that the authors have fully used the CONSORT guidance in reporting the trial results - a CONSORT flow diagram is included (fig 2) which is great, but the full CONSORT guidance and checklist should also be used - and the completed checklist uploaded with the revised paper as supporting information. The authors can use the regular CONSORT instruments (https://www.equator-network.org/reporting-guidelines/consort/) or the more recent specialist guideline for reporting multi-arm randomized trials (https://jamanetwork.com/journals/jama/fullarticle/2731183). 

*Per journal policy a copy of the final trial protocol should be uploaded as supporting information with the revised paper, or alternatively if the protocol was published in a journal then the citation for that can be given (in which case no need to upload the file).

*In the paper, a number of post-hoc analyses are reported, eg those regarding improvement of language scores in boys only. It would be good to include some further discussion about the possible limitations of such post-hoc (presumably, not specified in the protocol) analyses, and also to what extent those findings are biologically plausible (which would strengthen the claims). 

Comments from the reviewers:

Reviewer #1: Dear authors, kindly see my attached letter. Kindly keep up the good work. We as colleagues in this field want more information from this study. (see attached pdf)

Reviewer #2: Alex McConnachie, Statistical Review

Olsen and colleagues report a secondary analysis from a 2x2x3 factorial trial of food supplementation in Burkina Faso. This review considers the statistical aspects of the paper.

These are generally good. The outcome variables are validated measures, expressed as z-scores. The basic analysis plan involves fitting appropriately-adjusted regression models with 3-way intervention effect interactions and applying a backward selection procedure. The main effects and intervention interactions are reported clearly.

Subgroup analyses were applied to a number of baseline factors, though only interactions with sex are reported in the text, and in footnotes to the tables. None of the other factors indicated any intervention effect heterogeneity. Given that only sex showed any sign of interaction with the interventions, would a forest plot be a good way to present these more systematically?

Reviewer #3: The team from Burkina Faso continues their series of papers analyzing a very impressive set of data and findings from their 2x2x3 factorial RCT of a variety of different supplementary foods for MAM. This study has been the source of several high-profile papers in the malnutrition community. Not surprisingly, given this experience, the manuscript is very well written and presented; the methods are tight and the results are well-analyzed. I usually have many many comments when reviewing manuscripts, but here the authors have delivered an excellently written manuscript.

Here, they focus on the extremely important outcomes of developmental milestones among survivors. The authors are to be congratulated for their attention to cognitive and physical development, something that is all too often lost in our rush to just focus on survival and nutritional recovery. As they are implicitly (and maybe even explicitly) emphasizing, it is just as important to survive well as to survive at all.

It is reassuring (in terms of internal face validity) to see that there is some dose-response relationship between the percentage of milk protein consumed and developmental scores.

Any explanation for the sex-specific effects seen?

Some exploration of cost-benefit analysis would be helpful as well.

It would be fascinating in future research to evaluate these same developmental outcomes in trials where specific supplementation is not provided -- for example, those that are trialing direct cash transfer or counseling interventions.

[LINK]

---

## [Decision Letter · Decision Letter 2]

7 Oct 2020

Dear Dr. Olsen,

Thank you very much for re-submitting your manuscript "Effectiveness of food supplements on early child development in children with moderate acute malnutrition: a randomized 2 x 2 x 3 factorial trial in Burkina Faso" (PMEDICINE-D-20-02543R2) for review by PLOS Medicine.

I have discussed the paper with my colleagues and the academic editor and it was also seen again by two reviewers. I am pleased to say that provided the remaining editorial and production issues are dealt with we are planning to accept the paper for publication in the journal.

[LINK]

We look forward to receiving the revised manuscript by Oct 14 2020 11:59PM. 

Sincerely,

Artur Arikainen

Associate Editor

PLOS Medicine

plosmedicine.org

Requests from Editors:

1. Please implement the final reviewer comments.

2. Please update the title to: “Impact of food supplements on early child development in children with moderate acute malnutrition in Burkina Faso: neuropsychiatric outcomes from a randomized 2 x 2 x 3 factorial trial”.

3. The Data Availability Statement (DAS) requires revision. If the data are not freely available, please include an appropriate contact (web or email address) for inquiries (this cannot be a study author).

4. Abstract:

a. Around lines 31-32, please include trial recruitment dates, and mention that it “took place at 5 health centres in Province du Passoré, Northern Region”. 

b. Please state whether or not there was any blinding of participants or staff.

c. Around lines 33-34, please state the outcomes as follows: “This analysis presents secondary outcomes from the trial, which were…”.

d. Around line 35, please give the age and sex demographics of the participants. Please also state how many were randomized to each group, and how many completed treatment in each group.

e. Line 36: Change to: “At 12 weeks, mean MDAT z-scores in the whole trial cohort had increased…”

f. Line 38: We note that the 20% milk data are not statistically significant. Please reword to say that 50% was significant while 20% was not.

g. Lines 39-40: Please reword to “Fine motor z scores were improved in participants receiving ...”

h. Line 41: Please avoid the term “tended”; instead please simple state whether this difference is statistically significant or not.

i. Please quantify all results with p values and 95% CIs, including eg. lines 42-44.

j. Please use square brackets when nesting, eg. “…(…[…]…)…”

k. Please mention whether or not any adverse events were recorded, and whether they were attributed to treatment.

l. Line 46: Please begin with “In this study, we found that…”

m. Please include the trial registration number and primary source(s) of funding at the end.

5. Author Summary:

a. Line 59: Please clarify “factorial” to a lay reader, or rephrase more simply.

b. Line 63: Please clarify “cognitive z-scores” to a lay reader, or rephrase more simply.

6. Please leave a space between text and citations, eg: “…cognitive skills in children[1] and the…”

7. Line 124: Please include trial recruitment dates.

8. Line 166: Please confirm whether consent was “informed”.

9. Results: Please present the safety data for the study including numbers of specific events and whether or not adverse events are thought to be related to treatment.

10. Lines 316, 323, 326, 437: Please replace “marginally significant”/”marginally associated” with “not significant”/”not associated”.

11. Lines 346-347: Please reword to: “…appeared to be more effective…”

12. Line 348: Please avoid the term “tendency”; instead please simple state whether this difference is statistically significant or not.

13. Page 21: Please remove all sections here except the Acknowledgements.

14. Please provide a DOI or URL for references 14, 15, and 24.

15. Please remove “Lond Engl.” from references 1 and 2.

16. When completing the CONSORT checklist, please use section and paragraph numbers, rather than page numbers.

Comments from Reviewers:

Reviewer #1: Dear authors, well done, I have some suggestions which I add in a separate letter, regards, Ingunn 

Reviewer #3: The authors have satisfactorily answered my queries.

[LINK]

---

## [Editor Report · Decision Letter 3]

17 Nov 2020

Dear Dr Olsen, 

On behalf of my colleagues and the academic editor, Dr. James K Tumwine, I am delighted to inform you that your manuscript entitled "Impact of food supplements on early child development in children with moderate acute malnutrition: a randomised 2 x 2 x 3 factorial trial in Burkina Faso" (PMEDICINE-D-20-02543R3) has been accepted for publication in PLOS Medicine. 

PRODUCTION PROCESS

Before publication you will see the copyedited word document (within 5 business days) and a PDF proof shortly after that. The copyeditor will be in touch shortly before sending you the copyedited Word document. We will make some revisions at copyediting stage to conform to our general style, and for clarification. When you receive this version you should check and revise it very carefully, including figures, tables, references, and supporting information, because corrections at the next stage (proofs) will be strictly limited to (1) errors in author names or affiliations, (2) errors of scientific fact that would cause misunderstandings to readers, and (3) printer's (introduced) errors. Please return the copyedited file within 2 business days in order to ensure timely delivery of the PDF proof. 

If you are likely to be away when either this document or the proof is sent, please ensure we have contact information of a second person, as we will need you to respond quickly at each point. Given the disruptions resulting from the ongoing COVID-19 pandemic, there may be delays in the production process. We apologise in advance for any inconvenience caused and will do our best to minimize impact as far as possible.

EARLY VERSION

PRESS

PROFILE INFORMATION

Thank you again for submitting the manuscript to PLOS Medicine. We look forward to publishing it. 

Best wishes, 

Artur A. Arikainen, 

Senior Editor 

PLOS Medicine

plosmedicine.org